# Midbrain encodes sound detection behavior without auditory cortex

**Tai-Ying Lee, Yves Weissenberger, Andrew J King, Johannes C Dahmen***

Department of Physiology, Anatomy and Genetics, University of Oxford, Oxford, United Kingdom

## eLife assessment

This study demonstrates that neurons receiving inputs from auditory cortex in the inferior colliculus widely encode the outcome of a sound detection task independent of the presence of auditory cortex. This **valuable** study based on imaging of transsynaptically labelled neurons provides **convincing** evidence that auditory cortex is necessary neither for sound detection, nor to channel information related to behavioral outcome to the subcortical auditory system. This study will be of wide interest for sensory neuroscientists.

**\*For correspondence:**
johannes.dahmen@dpag.ox.ac.uk

## Abstract

Hearing involves analyzing the physical attributes of sounds and integrating the results of this analysis with other sensory, cognitive, and motor variables in order to guide adaptive behavior. The auditory cortex is considered crucial for the integration of acoustic and contextual information and is thought to share the resulting representations with subcortical auditory structures via its vast descending projections. By imaging cellular activity in the corticorecipient shell of the inferior colliculus of mice engaged in a sound detection task, we show that the majority of neurons encode information beyond the physical attributes of the stimulus and that the animals' behavior can be decoded from the activity of those neurons with a high degree of accuracy. Surprisingly, this was also the case in mice in which auditory cortical input to the midbrain had been removed by bilateral cortical lesions. This illustrates that subcortical auditory structures have access to a wealth of non-acoustic information and can, independently of the auditory cortex, carry much richer neural representations than previously thought.

## Introduction

Classically, perception is considered to rely on the flow of information from the sensory periphery via a sequence of hierarchically-organized brain structures up to the cortex. The ascending sensory pathways connecting these structures have been studied extensively and much has been learned about how signals are relayed, how features are extracted, and how information is integrated to produce increasingly abstract representations of the sensory environment. These pathways are paralleled by descending pathways that can feed information back to lower-order sensory structures. The fact that descending projections often outnumber their feedforward counterparts (*Sherman, 2007*) attests to their likely importance for brain function. This may include turning an otherwise passive, stimulus-driven device into an active and adaptive brain that is capable of processing sensory input within its behavioral context and, therefore, able to learn and create meaning (*Engel et al., 2001*; *Kraus and White-Schwoch, 2015*; *Malmierca et al., 2015*).

The descending projections of the auditory cortex target all major subcortical stations of the auditory pathway and are among the largest pathways of the brain (*Winer, 2005*; *Bajo and King, 2013*; *Antunes and Malmierca, 2021*), making them a particularly suitable system for investigating the

**eLife digest** Making sense of a sound and responding to it appropriately requires various parts of the nervous system to work together in a hierarchical and interconnected manner. For example, after the ear converts sound vibrations into electric signals, this information is sent to and pre-processed by the midbrain, a brain structure tasked with linking the auditory brain stem to sensory and motor systems. The signals are then relayed to the auditory cortex where they are further decoded and integrated with information emerging from other sensory and behavioral systems. These integrated auditory signals can then be fed back to the midbrain, potentially adjusting the signals delivered to its downstream targets.

Due to its integrative nature, neural activity in the auditory cortex is also shaped by non-acoustic input. Yet a growing body of evidence points to auditory neurons present in other regions than the cortex, such as the midbrain, being able to respond to non-acoustic information as well. It has typically been assumed that such responses are mediated by feedback from the auditory cortex.

To test this assumption, Lee et al. recorded the activity of auditory neurons in the midbrain of mice performing a sound detection task (that is, responding to a clicking sound by licking a waterspout). The analyses showed that most cells encoded not only basic sound properties (such as amplitude) but also information about the animal's behavioral response; in fact, the performance of an animal could be accurately inferred based on the activity patterns of such neurons. This was the case even in mice in which the auditory cortex had been removed, suggesting that the activity detected in the midbrain had not emerged due to cortical signals.

The findings by Lee et al. help refine our understanding of the brain processes that underpin hearing, in particular by highlighting tight links between behavioral information and neural activity in the midbrain. These results should help guide further research into how various brain regions participate in the processing of auditory input and the production of sound-guided behaviors, including when these mechanisms are affected by factors such as health or disease.

behavioral and physiological consequences of corticofugal processing. One of their main targets is the inferior colliculus (IC), an obligatory midbrain relay for nearly all ascending auditory input. The corticocollicular projection primarily terminates in the non-lemniscal shell of the IC. The shell encapsulates and is extensively connected with the central nucleus of the IC, which forms part of the tonotopically organized core or lemniscal auditory pathway to the primary auditory cortex. The projection from the auditory cortex to the midbrain was identified almost a century ago (*Mettler, 1935*) and decades of research have since demonstrated that manipulating the activity of descending projection neurons can alter the collicular representations of multiple sound features, influence adaptive plasticity and perceptual learning, and even trigger an innate flight response (*Suga, 2008*; *Nakamoto et al., 2008*; *Bajo et al., 2010*; *Xiong et al., 2015*; *Blackwell et al., 2020*). However, experimental evidence, especially from behaving animals, that could help explain what information the auditory midbrain and other subcortical sensory structures rely on their cortical input for is still very limited.

Interactions between different sensory pathways occur at multiple processing levels and they are also closely linked with the brain's motor centers and neuromodulatory regions. Indeed, recordings in awake animals have shown that behavior, cognition, and brain state can strongly influence activity in the sensory pathways (*Schneider and Mooney, 2018*; *McCormick et al., 2020*; *Parker et al., 2020*). Consistent with a hierarchical view of sensory processing in which neurons at higher levels carry progressively more complex representations of the world, such contextual influences appear particularly strong in the cortex (*Stringer et al., 2019*; *Musall et al., 2019*) and may to a large extent be the result of intracortical processing (*Noudoost et al., 2010*; *Schneider et al., 2014*; *Song et al., 2017*). Nevertheless, non-acoustic and contextual variables can also alter sensory processing at subcortical levels, including the IC and particularly its shell (*Metzger et al., 2006*; *Gruters and Groh, 2012*; *Chen and Song, 2019*; *Yang et al., 2020*; *Parras et al., 2017*; *Saderi et al., 2021*; *Shaheen et al., 2021*). This raises the possibility that these context-dependent effects may be inherited from the auditory cortex (*Ford et al., 2024*).

To test whether auditory midbrain neurons convey behaviorally-relevant signals that depend on descending cortical inputs, we imaged corticorecipient IC shell neurons in mice engaged in a sound

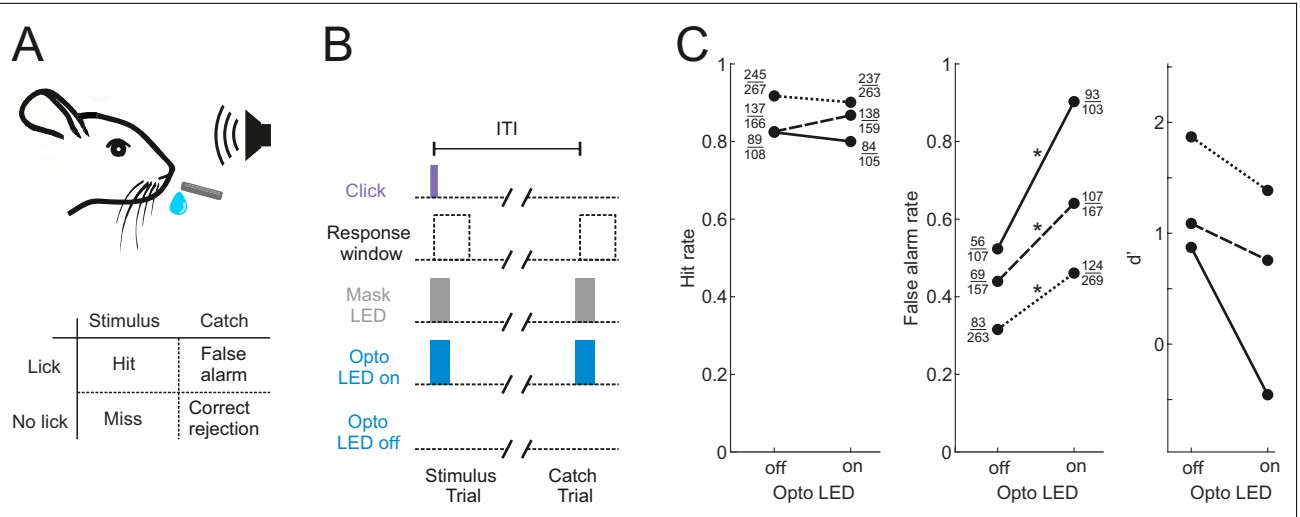

**Figure 1.** Optogenetic inactivation of the auditory cortex impairs sound detection performance in head-fixed mice. (**A**) Schematic of the click detection task. (**B**) Trial structure for experiments involving optogenetic manipulation. Stimulus trials (click) and catch trials (no click) were randomly interleaved and consecutive trials were separated by a randomly varying inter-trial interval (ITI). LEDs placed over each auditory cortex were switched on randomly in half of the stimulus and catch trials to photoactivate the opsin. A separate set of LEDs (Mask LEDs) placed directly in front of the mouse's eyes were switched on in all Opto-on and Opto-off trials to prevent mice from visually registering the light from the photoactivation LEDs. (**C**) Detection performance in trials during which light was shone on the auditory cortex for optogenetic silencing (Opto LED – on) vs control trials (Opto LED - off). Different line styles indicate different mice (n=3). Numbers next to data points indicate the numbers of hit and false alarm trials over a total number of stimulus and catch trials, respectively. *p<0.001, two-sided Chi-squared proportion test.

detection task. We found that the activity of most neurons contained information beyond the physical attributes of the sound and that this information could be used to decode the animals' behavior with a high degree of accuracy. Surprisingly, this was the case both in mice with an intact cortex and those in which the auditory cortex had been lesioned. These findings suggest that subcortical auditory structures have access to a wealth of non-auditory information independently of descending inputs from the auditory cortex. Consequently, the contextually-enriched representations that are characteristic of sensory cortices can arise from subcortical processing.

## Results

### Transient suppression of the auditory cortex impairs sound detection

Our aim was to characterize the activity of neurons in the shell of the IC in animals engaged in sound-guided behavior and assess how this activity is influenced by the input from the auditory cortex. To this end, we trained water-regulated mice on a sound detection task (*Figure 1A*) in which they were rewarded with a drop of water for licking in response to a click sound. Transient pharmacological silencing of the auditory cortex using the GABA-A agonist muscimol has been shown to abolish the ability of rodents (*Talwar et al., 2001*), including head-fixed mice (*Li et al., 2017*), to perform a sound detection task, making this approach unsuitable for our aim of exploring the role of IC during behavior. We found that optogenetic suppression of cortical activity by photoactivating ChR2-expressing inhibitory neurons in *Gad2-IRES-cre* mice (*Lohse et al., 2020*) also significantly impaired sound detection performance (*Figure 1B and C*), albeit not to the same degree as pharmacological silencing. Although a control group in which the auditory cortex was injected with an EYFP virus lacking ChR2 would be required to confirm that the altered behavior results from an opsin-dependent perturbation of cortical activity, this result shows that this manipulation is also unsuitable for our study as it would leave us unable to determine whether any changes in the activity of IC neurons arise from removal of their auditory cortical input or are a consequence of alterations in the animals' behavior.

## Auditory cortex lesions leave detection ability intact

Several recent studies have shown that in contrast to the disruptive effects of transient silencing, cortical lesions leave performance in some sensory tasks intact (*Hong et al., 2018*; *Ceballo et al., 2019*; *O'Sullivan et al., 2019*). In order to assess how auditory cortex lesions impact sound detection performance, we therefore compared the performance of mice with bilateral lesions of the auditory cortex (n=7) with non-lesioned controls (n=9).

Most corticocollicular neurons project ipsilaterally, with a substantial proportion also sending axons to the contralateral midbrain (*Stebbings et al., 2014*). The majority of corticocollicular neurons are found in the temporal cortex, and overwhelmingly in the auditory fields, while a small fraction populates adjacent areas, such as the temporal association area (*Figure 2—figure supplement 1*). After the experiments, we injected a retrogradely-transported viral tracer (rAAV2-retro-tdTomato) into the right IC to determine whether any corticocollicular neurons remained after the auditory cortex lesions (*Figure 2*, *Figure 2—figure supplement 2*, *Figure 2—figure supplement 3*). The presence of retrogradely-labeled corticocollicular neurons in non-temporal cortical areas (*Figure 2*) was not the result of viral leakage from the dorsal IC injection sites into the superior colliculus (*Figure 2—figure supplement 3*).

The ability of the mice to learn and perform the click detection task was evident in increasing hit rates and decreasing false alarm rates across training days (*Figure 3A*, p<0.01, mixed-design ANOVAs). There was no difference between lesioned and non-lesioned mice in their learning speed (*Figure 3A*, p>0.05, mixed-design ANOVAs) or psychometric functions (*Figure 3B*, p>0.05, mixed-design ANOVA). Cortical lesioning thus leaves behavioral sensitivity to clicks intact and therefore provides a means of examining the effects of removing corticocollicular input, albeit non-reversibly, without directly affecting sound detection performance.

## Transsynaptic labeling and two-photon calcium imaging of auditory corticorecipient IC neurons

Manipulations of auditory cortical activity can influence the activity of neurons throughout the IC, including the central nucleus (*Suga, 2008*; *Nakamoto et al., 2008*), where corticocollicular axons are relatively sparse (*Stebbings et al., 2014*). The strongest effects, however, tend to be observed in the shell, where cortical input is densest (*Nakamoto et al., 2008*; *Vila et al., 2019*; *Blackwell et al., 2020*). But even here, effects can be subtle (*Vila et al., 2019*) or undetectable (*Blackwell et al., 2020*), especially for cortical silencing. It is also unclear whether the IC neurons recorded in these studies receive cortical input or not. Therefore, we took a projection-specific approach to record the activity of IC neurons that receive direct input from the auditory cortex. More specifically, we injected AAV1.hSyn.Cre.WPRE, a virus with anterograde transsynaptic spread properties (*Zingg et al., 2017*), into the right auditory cortex of, initially, a tdTomato (Ai9) reporter mouse. This resulted in the expression of Cre recombinase and the reporter gene in neurons that receive input from the auditory cortex, including the corticorecipient neurons of the IC (*Figure 4A*). By employing this approach in GCaMP6f (Ai95D) reporter mice, we could target the expression of a calcium indicator to corticorecipient IC neurons. We then proceeded to record the activity of corticorecipient neurons within about 150 µm of the dorsal surface of the IC using two-photon microscopy (*Figure 4B*, *Figure 4—video 1*).

## Corticorecipient IC neurons display heterogeneous response profiles

The activity of individual corticorecipient IC neurons showed distinct response profiles across neurons and trial outcomes (hit vs miss) (*Figure 4C*). While averaging across all neurons cannot capture the diversity of responses, the averaged response profiles suggest that it is mostly trial outcome rather than the acoustic stimulus and neuronal sensitivity to sound level that shapes those responses (*Figure 4—figure supplement 1*). Indeed, close to half (1272/2649) of all neurons showed a statistically significant difference in response magnitude between hit and miss trials, while only a small fraction (97/2649) exhibited a significant response to the sound. While the number of sound-responsive neurons is low, this is not necessarily surprising given the moderate intensity and very short duration of the stimuli. For comparison: Using the same transgenics, labeling approach, and imaging setup and presenting 200 ms long pure tones at 60 dB SPL with frequencies between 2 kHz and 64 kHz, we typically find that between a quarter and a third of neurons in a given imaging area exhibit a statistically significant response (data not shown).

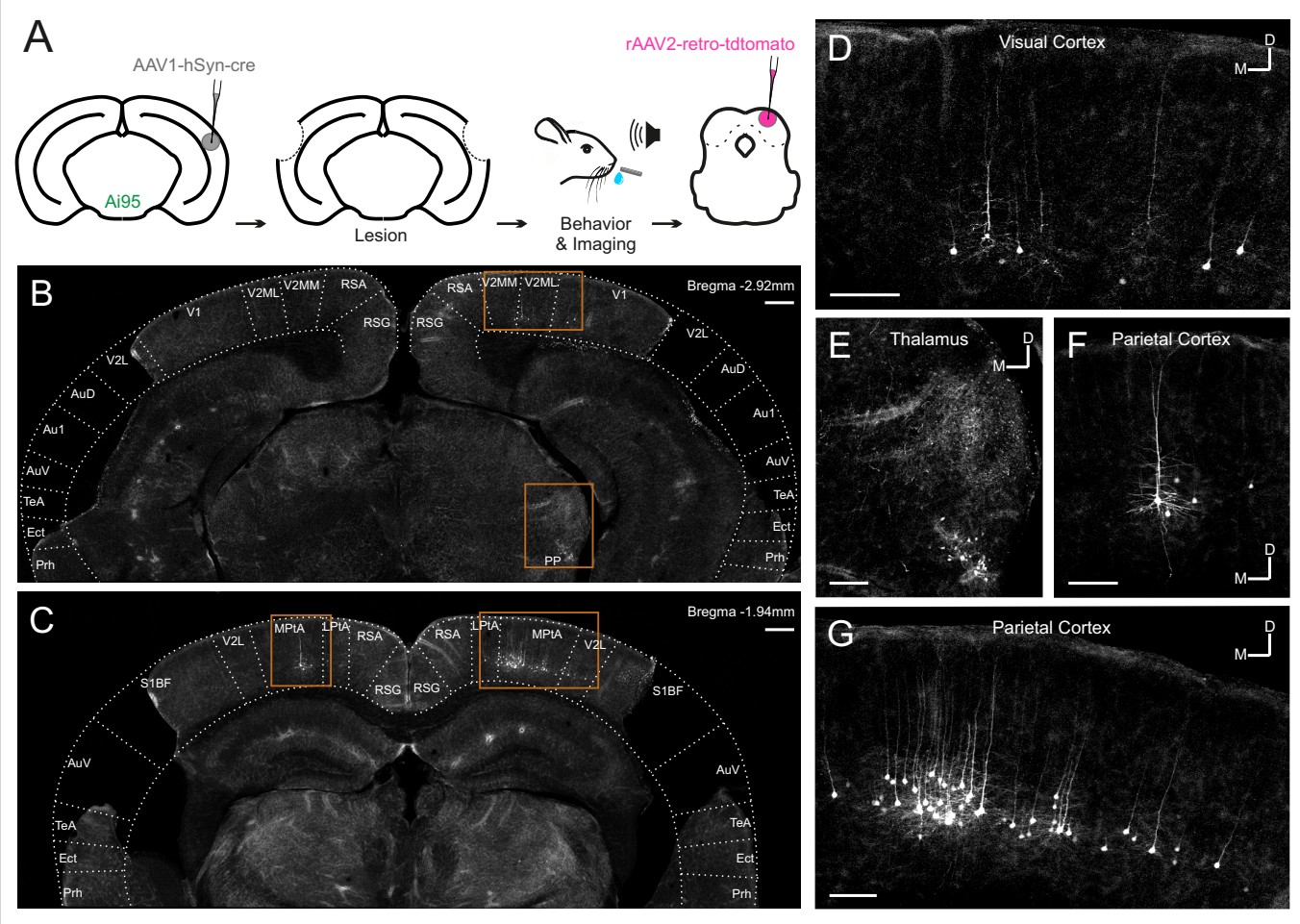

**Figure 2.** Retrograde viral tracing of inferior colliculus (IC)-projecting neurons in bilaterally lesioned mice. (**A**) Timeline of experimental procedures. AAV1.hSyn.cre.WPRE was injected into the right auditory cortex of GCaMP6f-reporter (Ai95D) mice. This causes transsynaptic delivery of the virus to the IC and expression of GCaMP6f in corticorecipient IC neurons. Several weeks later, the mice underwent bilateral lesioning of the auditory cortex either by aspiration or by thermocoagulation (see *Figure 2—figure supplement 2* for histological sections from a mouse that underwent thermocoagulation) and were implanted with a glass window over the right auditory cortex. Following recovery from this procedure, water access was restricted and, 2–3 days later, behavioral training and imaging commenced. After data collection had been completed, rAAV2-retro-tdTomato was injected in the dorsal IC in order to label corticocollicular neurons that had remained intact. (**B, C**) Coronal sections showing lesion extent at different rostrocaudal positions for one example mouse. Area borders were drawn onto the images according to *Paxinos et al., 2001*. No retrogradely-labeled neurons were found near the lesion borders, suggesting that the auditory cortex had been completely removed. Corticocollicular projections from non-temporal regions as well as thalamocollicular projections remained intact. Scale bars, 200 μm. (**D**) High magnification image (location shown by the upper rectangle in (**B**)) showing corticocollicular neurons in the visual cortex. Scale bar, 100 μm. (**E**) High magnification image (location shown by the lower rectangle in (**B**)) showing thalamocollicular neurons in the peripeduncular nucleus of the thalamus (PP). Scale bar, 100 μm. (**F, G**) High magnification images (locations shown by the left and right rectangles in (**C**), respectively) showing corticocollicular neurons in the parietal cortex. Scale bars,100 μm. Cortical area abbreviations: Au1, primary auditory; AuD, secondary auditory, dorsal; AuV, secondary auditory, ventral; Ect, ectorhinal; LPta, lateral parietal association; MPta, medial parietal association; Prh, perirhinal; RSG, retrosplenial granular; RSA, retrosplenial agranular; S1BF, primary somatosensory, barrel field; TeA, temporal association; V1, primary visual; V2L, secondary visual, lateral; V2ML, secondary visual, mediolateral; V2MM, secondary visual, mediomedial.

The online version of this article includes the following figure supplement(s) for figure 2:

**Figure supplement 1.** Contra- and ipsilateral corticocollicular neurons along the rostrocaudal axis.

**Figure supplement 2.** Lesioning by thermocoagulation.

**Figure supplement 3.** Retrograde labeling of corticocollicular neurons in non-temporal areas of the cerebral cortex is not the result of viral leakage into the superior colliculus.

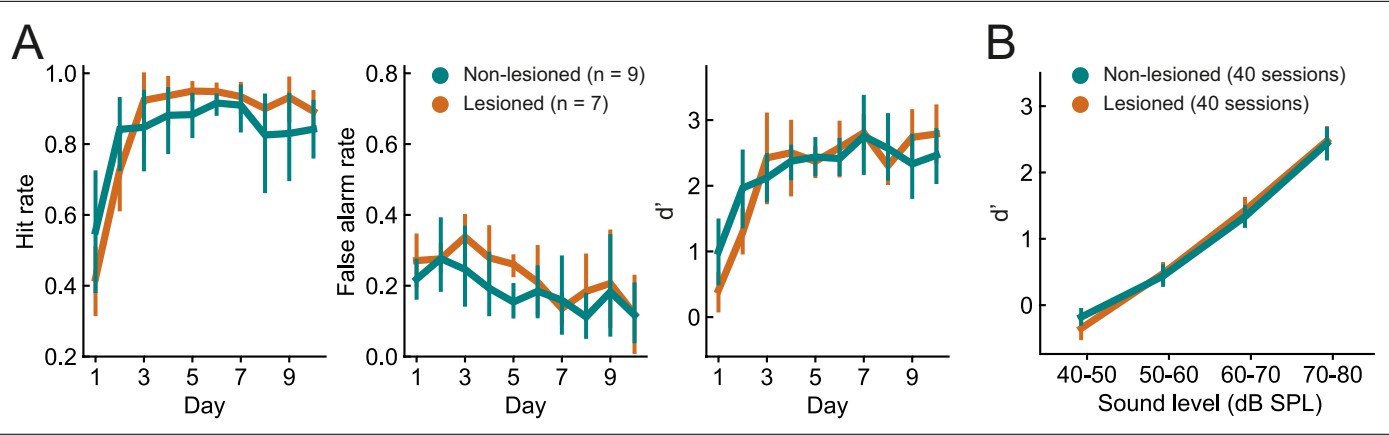

**Figure 3.** Lesioned and non-lesioned mice are indistinguishable in their click detection learning rate and sensitivity. (**A**) Hit rate, false alarm rate, and d' over time for lesioned and non-lesioned animals. (**B**) d' as a function of sound level. The sound levels used were not identical across all mice and were, therefore, combined into 10 dB wide bins. Error bars indicate 95% confidence intervals.

To capture the heterogeneity of response patterns across all recorded neurons, we used an unsupervised clustering algorithm (*Namboodiri et al., 2019*) to group the average responses on hit and miss trials for each neuron. This yielded 10 clusters that displayed different response patterns over the course of the trial (*Figure 5A and B*). Most of the clusters exhibited distinct activity for hit vs miss trials. Some hit trial profiles were characterized by increases or decreases in activity, with a very sharp, short-latency onset, as in clusters 4 and 10 (see *Figure 5—figure supplement 1* for a scaled version of cluster 10), and others by much more gradual changes in which a peak occurred seconds after the trial onset, as in clusters 5 and 9. Cluster 3, which contained the smallest number of neurons, was an exception in that it showed a transient, short latency response to the stimulus for both trial outcomes. The response profiles of some other clusters, especially clusters 6 and 8, were also qualitatively similar across hit and miss trials and/or only weakly modulated across both trial types.

This suggests that the activity of the majority of neurons in the recorded population contained information beyond the physical properties of the stimulus. Given that licking causes self-generated sounds, IC neurons could, in principle, respond to the sound of licking. However, given how quiet these are - estimated to be just 12 dB SPL (*Singla et al., 2017*) - and that much of the response to such lick-related sounds is already canceled out at the level of the cochlear nucleus (*Singla et al., 2017*; but see *Shaheen et al., 2021*), it is highly unlikely that lick-related sounds play a major role in driving activity in the IC.

To assess whether certain response profiles depended on auditory cortical input, we compared the ratio of neurons from lesioned vs non-lesioned mice in each cluster to that of the overall recorded population. The number of recorded neurons was unequal for lesioned and non-lesioned mice (952 vs 1697, respectively), reflecting the fact that a greater proportion of imaging sessions in non-lesioned animals were carried out using a larger field of view, which contained larger numbers of neurons (*Figure 5—figure supplement 2*). To account for this, the percentages shown on the pie charts were normalized to the ratio of the overall population (*Figure 5C*). Neurons from both groups were well represented across all 10 clusters and while a significant difference in the lesioned/non-lesioned ratio was found for four clusters, the difference between the groups was greater than 20% for only one of them. Furthermore, there was a close correspondence between the cluster averages of lesioned and non-lesioned mice (*Figure 5—figure supplement 3*). This suggests that the IC shell can produce very similar output regardless of whether auditory cortical input is available or not.

## Behavior can be accurately decoded from neural activity in lesioned and non-lesioned mice

The average responses of individual neurons in the IC shell exhibited a variety of activity patterns associated with both the stimulus and the trial outcome (*Figure 5A and B*). To gain insight into how these activity patterns can be read out collectively on a trial-by-trial basis, we assessed the relationship between the trial-by-trial network activity and the trial outcome. We trained logistic regression

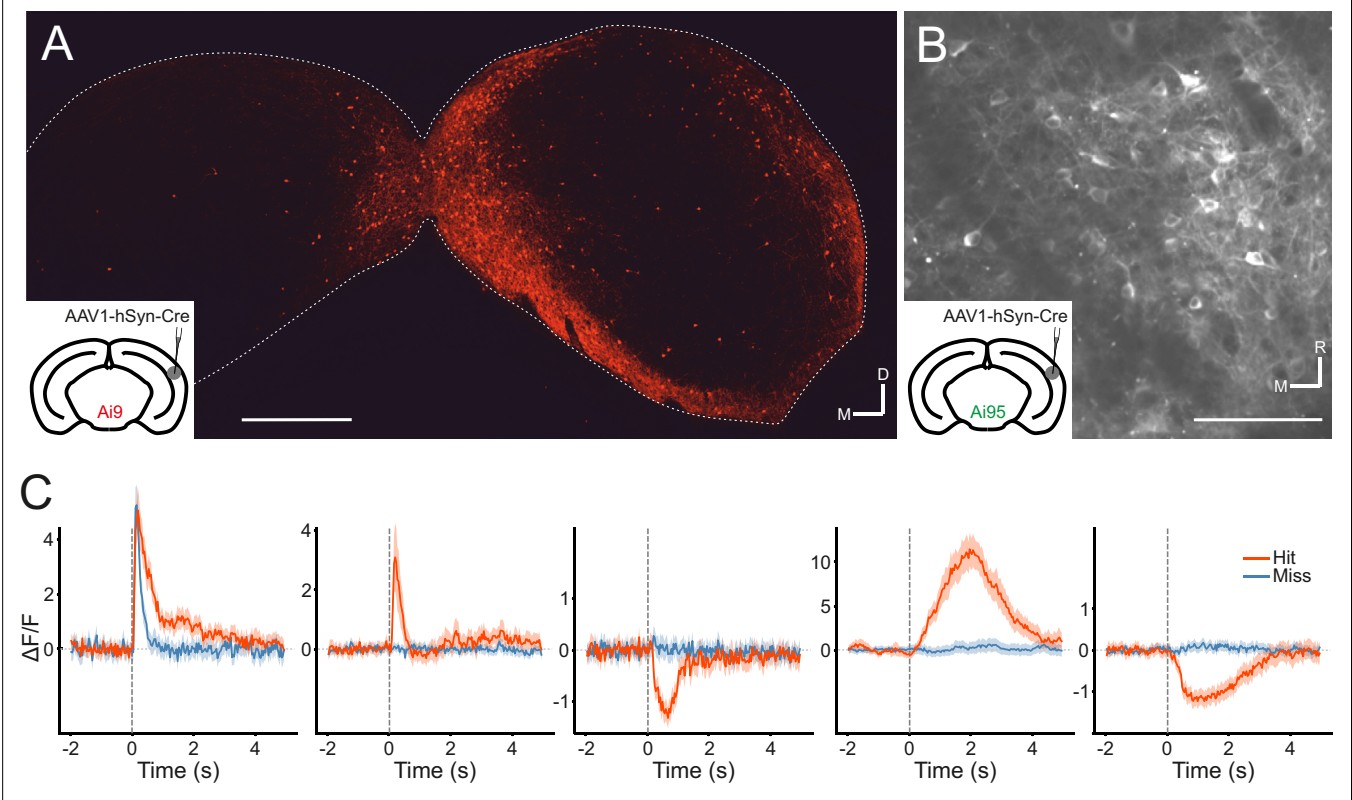

**Figure 4.** Transsynaptic targeting and two-photon calcium imaging of corticorecipient inferior colliculus (IC) shell neurons. (**A**) Coronal section of the left and right IC of a tdTomato-reporter (Ai9) mouse in which AAV1.hSyn.Cre.WPRE had been injected into the right auditory cortex three weeks before perfusion. The transsynaptically transported virus drove the expression of Cre recombinase and tdTomato in neurons that receive input from the auditory cortex, including the corticorecipient neurons in the IC. tdTomato-labeled neurons were predominantly found in the shell of the ipsilateral (right) IC. Scale bar, 500 µm. (**B**) In vivo two-photon micrograph taken approximately 100 µm below the dorsal surface of the right IC of a GCaMP6f-reporter mouse (Ai95D) in which GCaMP6f expression had been driven in corticorecipient IC neurons by injection of AAV1.hSyn.Cre.WPRE into the right auditory cortex. See *Figure 4—video 1* for the corresponding video recording. Scale bar, 100 µm. (**C**) Example average response profiles of five corticorecipient IC neurons for different trial outcomes. Vertical line at time 0 s indicates the time of click presentation. Shaded areas represent 95% confidence intervals.

The online version of this article includes the following video and figure supplement(s) for figure 4:

**Figure supplement 1.** Averaged response profiles for stimulus and catch trials.

**Figure 4—video 1.** Two-photon calcium imaging was performed approximately 100 µm below the dorsal surface of the right inferior colliculus (IC) of a GCaMP6f-reporter mouse (Ai95D) engaged in a sound detection task.

https://elifesciences.org/articles/89950/figures#fig4video1

models to classify hit vs miss trials on a trial-by-trial, frame-by-frame basis. As different populations of neurons were recorded in different imaging sessions, the models were trained separately for each session. 'Dummy models,' which randomly classified trials while taking into account the probability of hit vs miss trials in a given session, were used as the baseline model performance. If the population activity of the IC shell contained information about the trial outcome, the performance of the models would be significantly above baseline.

In both lesioned and non-lesioned mice, the average model performance was significantly above baseline in classifying hit vs miss trials ($p<0.05$, one-sided Wilcoxon signed-rank test or paired t-test with Bonferroni correction, *Figure 6A*), showed a temporal profile that is consistent with the dynamics of the activity profiles of some of the clusters, in particular clusters 1, 2, 4, 5, 9, 10 (*Figure 5A and B*), and was not meaningfully affected by differences in sound level distributions between hit and miss trials (*Figure 6—figure supplement 1*). Additionally, the model performance in non-lesioned mice was significantly better than that in lesioned mice ($p<0.05$, one-sided Mann-Whitney U test or t-test with Bonferroni correction, *Figure 6A*). This difference in the decoding performance was not the result

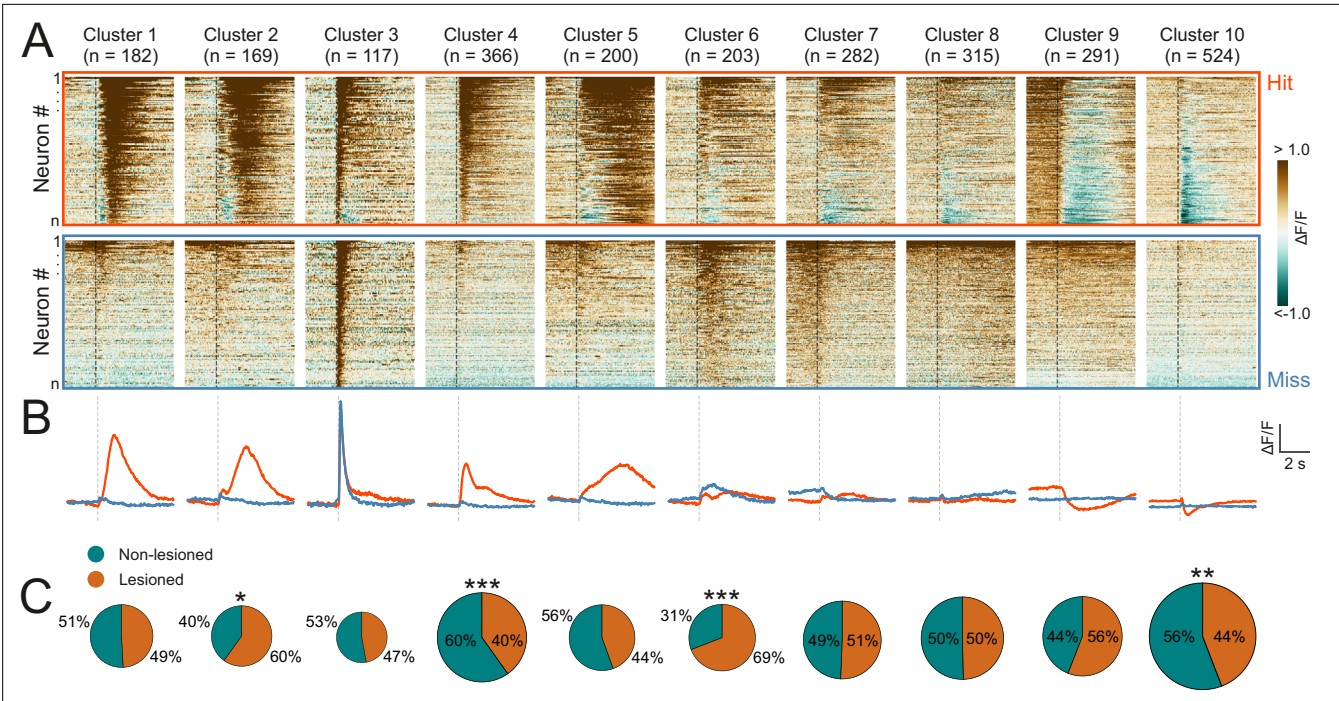

**Figure 5.** Corticorecipient inferior colliculus (IC) neurons display heterogeneous response profiles. (**A**) Peri-stimulus time histograms for all neurons in the dataset separated by cluster identity: hit trials (top) vs miss trials (bottom). (**B**) Averaged response profiles obtained by taking the mean across all neurons in a cluster separately for hit (red) and miss (blue) trials. (**C**) Pie charts illustrate the proportion of neurons from lesioned and non-lesioned mice in each cluster. The size of each pie chart is proportional to the total number of neurons in each cluster. Given the unequal number of neurons from lesioned (952 neurons) and non-lesioned (1697 neurons) mice, the pie charts were normalized to the overall sample size such that a 50/50 split indicates a lesioned/non-lesioned distribution that is identical to that of the overall population. Asterisks indicate a significant difference between the lesioned/non-lesioned distribution in the given cluster and that in the overall population. *p<0.05, **p<0.01, ***p<0.001, two-sided one proportion Z-test.

The online version of this article includes the following figure supplement(s) for figure 5:

**Figure supplement 1.** Rescaled response profiles for each cluster.

**Figure supplement 2.** Number of sessions for each imaging field of view size.

**Figure supplement 3.** High correspondence between cluster profiles of lesioned and non-lesioned mice.

of the difference in the number of neurons between non-lesioned and lesioned mice (*Figure 6—figure supplement 2*).

By examining the corticocollicular labeling and referencing the histological sections against a mouse brain atlas (*Paxinos et al., 2001*), we categorized the mice according to lesion size. Four of the seven lesioned animals had '(near-)complete' lesions, meaning that all (*Figure 2*) or an estimated ~95% (*Figure 2—figure supplement 2*) of the auditory cortex had been lesioned, while the remaining mice had 'partial' lesions, with an estimated 15–25% of the auditory cortex left intact. To assess whether the size of the lesions impacted the decoding performance, we compared the model performance between mice that had (near-)complete lesions and mice that had partial lesions. This revealed that the average decoding performance for mice with (near-)complete lesions was significantly better than that measured for mice with partial lesions. While this pattern of results may be unexpected, it is consistent with work showing smaller lesions being associated with greater somatosensory processing deficits (*Hong et al., 2018*). Additionally, the decoding performance in mice with (near-)complete lesions was largely indistinguishable from that in mice with an intact auditory cortex. Although the proportion of individual neurons with distinct response magnitudes in hit and miss trials in lesioned mice did not differ from that in non-lesioned mice, it was significantly lower when separating out mice with partial lesions (*Figure 6—figure supplement 3*). These results imply that the activity of IC shell neurons can contain similar amounts of information about the animal's behavior regardless of whether descending input from the cortex is available or not (*Figure 6B*).

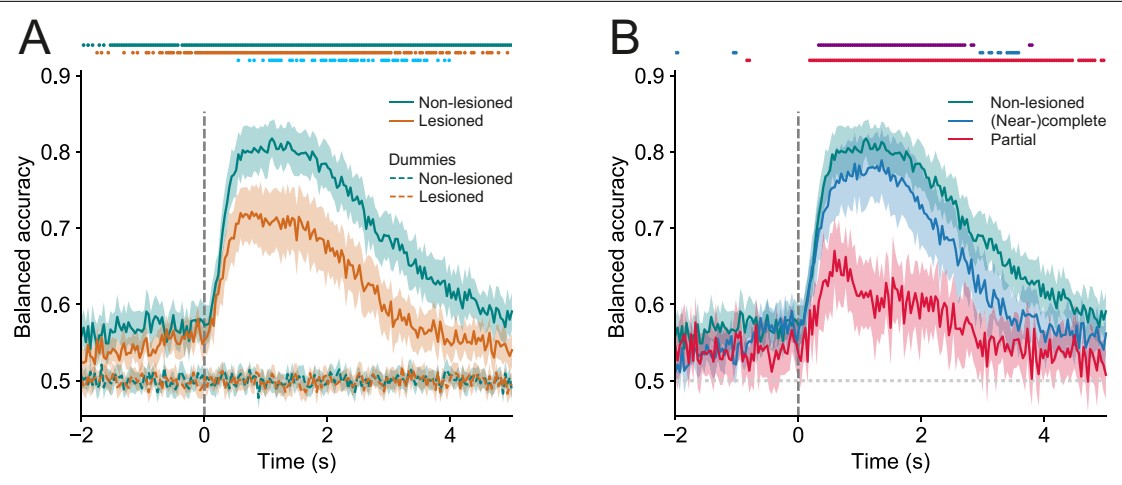

**Figure 6.** Trial outcomes can be accurately decoded from neural activity in lesioned and non-lesioned mice. (**A**) Average decoding accuracy of logistic regression models as a function of time against dummy models with a score of 0.5 meaning chance performance and a score of 1 being the maximum. Data shown depict the mean model accuracy across 37 (lesioned) and 38 (non-lesioned) sessions, respectively. Dots at the top indicate the time points (frames) where the model performance was significantly different between trained and dummy models for non-lesioned mice (teal) or lesioned mice (orange) ($p < 0.05$, one-sided Wilcoxon signed-rank test or paired t-test with Bonferroni correction, depending on whether normality assumption was met), and between the trained models for non-lesioned vs lesioned mice (blue) ($p < 0.05$, one-sided Mann-Whitney U test or t-test with Bonferroni correction, depending on whether normality assumption was met). (**B**) Same as (**A**) but the average model accuracy is plotted separately for mice with (near-)complete (22 sessions) and partial lesions (15 sessions). Dots at the top indicate the time points where the model performance was significantly different between partial vs (near-)complete mice (purple), (near-)complete vs non-lesioned mice (blue), and partial vs non-lesioned mice (red) ($p < 0.05$, one-sided Mann-Whitney U test or t-test with Bonferroni correction, depending on whether normality assumption was met). Shaded areas represent 95% confidence intervals.

The online version of this article includes the following figure supplement(s) for figure 6:

**Figure supplement 1.** Trial outcome decoding is not meaningfully affected by differences in sound level distributions between hit and miss trials.

**Figure supplement 2.** Greater number of recorded neurons was not associated with better decoding performance.

**Figure supplement 3.** Similar fractions of task-modulated and sound-driven neurons in lesioned and non-lesioned mice.

**Figure supplement 4.** Lick rates in peri-catch trial periods approximate next-trial-probability.

## Pre-stimulus activity is predictive of the upcoming trial outcome

Remarkably, decoding accuracy was better than baseline even before stimulus onset. This could reflect changes in the network state that led or contributed to the upcoming trial outcome. For instance, changes in arousal or motivation can alter both the probability that an upcoming stimulus is detected and the activity of neurons in the network (*Lee and Dan, 2012*; *McGinley et al., 2015*). The decoding models might detect such changes in activity, resulting in higher decoding accuracy prior to stimulus onset. Additionally, pre-stimulus differences in hit and miss trial activity could also reflect the anticipation of an upcoming stimulus (*Ruth et al., 1974*; *Nienhuis and Olds, 1978*; *Metzger et al., 2006*) and the resulting change in attentional state. Inter-trial intervals in our experiments were randomly drawn from a normal distribution with a mean and standard deviation of 8 s and 2 s, respectively, and a lower bound of 3 s. Nevertheless, spontaneous licks did not occur at random times during the peri-catch trial periods following hit trials. Instead, average lick rates approximated the inter-trial interval distribution (*Figure 6—figure supplement 4A–D*), suggesting that mice learned to adapt their behavior to this distribution and anticipate the timing of upcoming stimuli (*Figure 6—figure supplement 4E and F*). Assuming that successfully anticipating the timing of an upcoming stimulus confers a greater chance of detecting the stimulus, neurons whose activity reflects that anticipation might be expected to show differences in pre-trial activity between hit and miss trials that could be detected by a decoding model. Note that for the analysis illustrated in *Figures 5 and 6*, hit trials were excluded if there were any licks between –500 ms and +120 ms (the latter number representing the lower bound of the animals' lick-latency) relative to stimulus onset, suggesting that changes in pre-stimulus activity cannot be directly related to licking.

## Discussion

Imaging auditory corticorecipient neurons in the dorsal shell of the IC in mice trained to perform a sound detection task revealed that the majority of neurons exhibited distinct activity profiles for hit and miss trials, implying that they encode information beyond just the physical attributes of the stimulus. Indeed, using logistic regression models to classify hit vs miss trials, we found that the animals' behavioral choices can be read out from these neurons with a high degree of accuracy. Importantly, the difference in IC activity between hit and miss trials was observed across different sound levels and was not due to a difference in the sound level distribution for these two trial outcomes. Surprisingly, neural activity profiles and the decoding performance were similar in mice in which the auditory cortex had been lesioned bilaterally, suggesting that the midbrain has, independently of the auditory cortex, access to a wealth of non-acoustic information, which may be sufficient to support sound detection behavior.

Auditory corticocollicular axons terminate predominantly in the shell of the IC (*Stebbings et al., 2014*; *Bajo and King, 2013*) and the strongest effects of cortical manipulations have been reported in this region (*Nakamoto et al., 2008*; *Vila et al., 2019*; *Blackwell et al., 2020*). However, these effects can be subtle (*Cruces-Solís et al., 2018*; *Vila et al., 2019*) or undetectable, especially when optogenetic silencing is used (*Blackwell et al., 2020*). Because of this and uncertainties over exactly what proportion of neurons in the shell of the IC is innervated by the auditory cortex and even where the border lies with the underlying central nucleus (*Barnstedt et al., 2015*), we used an anterograde transsynaptic tagging approach (*Zingg et al., 2017*) to identify corticorecipient neurons. This, therefore, maximized the chances of revealing the contribution of descending cortical input to the response properties of these midbrain neurons. We imaged across the optically accessible dorsal surface of the IC down to a depth of about 150 μm below the surface. Consequently, the neurons we recorded were located predominantly in the dorsal cortex. However, identifying the borders between different subdivisions of the IC is not straightforward and we cannot rule out the possibility that some were located in the lateral cortex.

### Inferior colliculus neurons exhibit task-related activity

Our recordings from corticorecipient neurons in the IC are consistent with previous studies demonstrating that neural representations of behavioral variables can be found in the auditory midbrain (*Ruth et al., 1974*; *Nienhuis and Olds, 1978*; *Metzger et al., 2006*; *Gruters and Groh, 2012*; *Chen and Song, 2019*; *Yang et al., 2020*; *Saderi et al., 2021*; *De Franceschi and Barkat, 2021*; *Shaheen et al., 2021*; *Quass et al., 2024*). In keeping with responses recorded in the auditory cortex (*Francis et al., 2018*; *De Franceschi and Barkat, 2021*) and IC (*Chen and Song, 2019*; *Yang et al., 2020*; *De Franceschi and Barkat, 2021*) of behaving mice, we found that the activity of most neurons was facilitated and about a third were suppressed during the sound detection task. Overall, only a small minority of clusters (mostly cluster 3) in our dataset showed what could be characterized as largely behavior-invariant response profiles to the auditory stimulus. In contrast, a large number of neurons were clearly driven by variables other than the stimulus itself. Their activity may represent the choice (to lick or not to lick) that an animal made, preparatory motor activity, corollary discharge, or the reward and the somatosensory or gustatory feedback associated with its consumption, as well as modulation by the animal's cognitive and behavioral state. Due to the task structure used, for the most part, it was not possible to unambiguously assign activity profiles to a particular variable. Nevertheless, we can speculate that neurons with late transients, such as in cluster 5, are more likely to represent corollary discharge and signals associated with the consumption of the reward, while those with very short latency peaks, as in clusters 4 and 10, may represent the animals' choice and/or preparatory motor activity.

When engaged in the detection task, an animal's arousal or motivational state may vary spontaneously or as a result of changes in, for instance, thirst, time of day, or time into a session. In addition, cognitive factors, such as expectations about the timing of an upcoming trial (*Ruth et al., 1974*; *Nienhuis and Olds, 1978*; *Metzger et al., 2006*), which mice may have derived by learning the shape of the inter-trial interval distribution, may lead to variations in arousal or attentional state. Pre-trial differences in activity as well as the above-chance decoding performance before trial onset likely reflect the joint impact of those state changes on the activity of IC corticorecipient neurons and detection sensitivity (*McCormick et al., 2020*).

# Contribution of the auditory cortex to task-related activity in the midbrain

Given the massive corticofugal projections that exist within the auditory system (*Bajo and King, 2013*), we hypothesized that task-related activity in the IC might depend on descending inputs from the auditory cortex. To address this, we imaged corticorecipient IC neurons during the same sound detection task after removing the cortical input. Consistent with previous work in the auditory (*O'Sullivan et al., 2019*) and somatosensory systems (*Hong et al., 2018*), we found that transient optogenetic silencing of the auditory cortex impaired sound detection, whereas cortical lesions had no effect on detection behavior, with lesioned mice learning the task as quickly as non-lesioned animals and achieving the same level of performance. In order to determine whether the absence of auditory cortical input alters the activity of IC neurons during sound detection behavior, we therefore focused on mice with bilateral cortical lesions to avoid the potentially confounding effects that reduced detection sensitivity produced by transient cortical silencing might have on the activity of IC neurons. For the same reason, we opted against the more targeted approach of optogenetic silencing of corticocollicular axons. Furthermore, it would have been difficult to silence the entire corticocollicular projection and the higher light powers required for manipulating axons compared to somata would have risked transmitting light to the cortex or other corticofugal targets, potentially causing behavioral changes and/or sacrificing specificity. Locally silencing corticocollicular axons would also have left indirect transmission via the thalamus between the auditory cortex and IC intact and would have been very challenging to verify. Finally, it has been reported that using optogenetic silencing tools in axons can have unintended consequences (*Wiegert et al., 2017*).

In keeping with our findings, numerous studies (reviewed in e.g. *Pickles, 1988*; *Buser and Imbert, 1992*) have shown that simple auditory skills, including the ability of freely moving rats to detect sounds (*Kelly, 1970*), are unaffected by the removal of the auditory cortex. However, transient pharmacological silencing of the auditory cortex in freely moving rats (*Talwar et al., 2001*), as well as head-fixed mice (*Li et al., 2017*), completely abolishes sound detection (but see *Gimenez et al., 2015*). The time course of the effects produced by muscimol application (*Talwar et al., 2001*) suggests that there is a relationship between the size of the behavioral deficit and the degree of cortical inactivation. Consequently, milder impairments may be produced by the optogenetic approaches employed by us and others (*Kato et al., 2015*; *O'Sullivan et al., 2019*) because of incomplete suppression of cortical activity. Alternatively, the larger behavioral effects reported following muscimol application may be due to diffusion of the drug to other brain structures, potentially including the IC. Although our results cannot speak directly to the question of whether the preservation of sound detection without auditory cortex reflects a rewiring or repurposing of circuits in the brain, this seems unlikely given that other studies have shown that trained mice achieve pre-lesion performance levels on simple auditory discrimination (*Ceballo et al., 2019*; *O'Sullivan et al., 2019*) or somatosensory detection (*Hong et al., 2018*) tasks suddenly and within 48 hr following cortical ablation.

Why then does transient inactivation produce behavioral deficits? One possibility is that disabling the auditory cortex impacts behavior not because it contributes necessary computations or information, but because of the sudden and disruptive removal of tonic excitation (*Oberle et al., 2022*) to downstream targets (*Otchy et al., 2015*) that are indispensable for successful sound detection. In this scenario, normal operation would resume once synaptic scaling (*Keck et al., 2013*) had homeostatically restored normal activity in these structures, a process that has been suggested to take up to 48 hr and is consistent with the time course of recovery after lesions (*Ceballo et al., 2019*; *Hong et al., 2018*). Alternatively, several circuits may redundantly support sound detection. Silencing the auditory cortex might then transiently impede sound detection until the relevant downstream decision and motor structures have updated their synaptic weights and/or processing has shifted to the other circuits. Two observations, however, argue against this possibility. First, removing one of several redundant structures should leave some residual function intact and not have the devastating effect that pharmacological cortical silencing achieves (*Talwar et al., 2001*, *Li et al., 2017*). Second, other circuits mediating the acousticomotor transformation required for successful sound detection behavior very likely incorporate subcortical auditory structures, including the auditory midbrain. Activity in the IC may trigger actions (*Casseday and Covey, 1996*), such as licking, via its direct projections to the superior colliculus, pontine nuclei and the periaqueductal gray (*Huffman and Henson, 1990*; *Wenstrup et al., 1994*; *Casseday and Covey, 1996*; *Xiong et al., 2015*) or indirectly via its projections to the

auditory thalamus. If cortical lesioning results in a greater weight being placed on the activity in spared subcortical circuits for perceptual judgements, we would expect the accuracy with which trial-by-trial outcomes could be read out from IC neurons to be greater in mice without auditory cortex. However, that was not the case. This could imply that, following cortical lesions, greater weight is placed on structures other than the IC, with the thalamus being an obvious candidate, or that the auditory midbrain, thalamus and cortex are bypassed entirely if simple acousticomotor transformations, such as licking a spout in response to a sound, are handled by circuits linking the auditory brainstem and motor thalamus via pedunculopontine and midbrain reticular nuclei (*Inagaki et al., 2022*).

Some differences were observed for mice with only partial lesions of the auditory cortex. Those mice had a lower proportion of neurons with distinct response magnitudes in hit and miss trials than mice with (near-)complete lesions. Furthermore, trial outcomes could be read out with lower accuracy from these mice. While this finding is somewhat counterintuitive and is based on only three mice with partial lesions, it has been observed before that smaller lesions can have a more disruptive effect than larger, more complete lesions, in that the time it takes mice to learn a whisker-dependent sensory detection task is anticorrelated with the size of their somatosensory cortex lesion (*Hong et al., 2018*). While the complete destruction of a cortical area severs all its communication with downstream structures, a partial lesion may actually be more disruptive by eradicating normal local processing while at the same time leaving intact some tissue, especially in the deeper output layers, which continues to transmit what are now aberrant activity patterns. The difference in decoding accuracy that we observed in the IC could thus be a consequence of residual and now disruptive cortical input.

Our results show that behavioral variables are encoded by corticorecipient neurons in the dorsal shell of the IC independently of their main source of descending input, the auditory cortex. It therefore seems likely that this region of the auditory midbrain is part of the circuit that supports sound detection behavior in the absence of the auditory cortex. Nevertheless, except for the regions immediately bordering the auditory cortex, corticocollicular neurons located in other areas were left intact. These relatively sparse descending projections to the IC, such as those originating from somatosensory cortical areas (*Lohse et al., 2020*; *Lesicko et al., 2016*) and the parietal cortex, may have contributed to the response profiles that we observed. Additional non-acoustic sensory input can reach the IC via brainstem nuclei (*Lesicko et al., 2016*; *Shore and Zhou, 2006*) and the superior colliculus (*Chen et al., 2018*; *Coleman and Clerici, 1987*). The latter, together with input from the substantia nigra (*Olazabal and Moore, 1989*) and the globus pallidus (*Moriizumi and Hattori, 1991*) may also be a source of motor signals, while state changes may impact on the IC via inputs from neuromodulatory structures, including the locus coeruleus and the subparafascicular, dorsal raphe and tegmental nuclei (*Chen and Song, 2019*, *Liu et al., 2023*).

## Conclusion

Behavior is a major determinant of activity in the non-lemniscal auditory midbrain and thus key to understanding how it contributes to hearing. The anatomical feature that defines this structure more than any others is its connection with the auditory cortex. While modulation of IC activity by this descending projection has been implicated in various functions, most notably in the plasticity of auditory processing, we have shown in mice performing a sound detection task that IC neurons show task-related activity in the absence of auditory cortical input. These results, therefore, emphasize more than ever the need to factor in subcortical processing when considering how the cortex contributes to sound-guided behavior.

## Materials and methods
### Animals

All experiments were approved by the Committee on Animal Care and Ethical Review at the University of Oxford and were licensed by the UK Home Office (Animal Scientific Procedures Act, 1986, amended in 2012). We used 22 (three female, 19 male) B6;129S-*Gt(ROSA)26Sor*$^{tm95.1(CAG-GCaMP6f)Hze}$/J (Ai95D, JAX 024105, Jackson Laboratories, USA), three (one female, two male) Gad2$^{tm2(cre)Zjh}$/J (JAX 010802), six female B6.Cg-*Gt(ROSA)26Sor*$^{tm9(CAG-tdTomato)Hze}$/J (Ai9, JAX 007909), two female Ai95D X Slc32a1$^{tm2(cre)}$$^{Lowl}$/J (JAX 016962), three female Ai95D X B6.Cg-Tg(*Camk2a*-cre)T29-1Stl/J (JAX 005359, Jackson Laboratories, USA), and three (one male, two female) C57BL/6NTac.*Cdh23*$^{753A>G}$ (MRC Harwell, UK)

mice. All mice were 9–15 weeks old during data collection. They were maintained on a 12 hr light/dark cycle and were housed at 20–24°C with a relative humidity of 45–65%.

## Surgeries

For all surgical procedures, mice were premedicated with intraperitoneal injections of dexamethasone (Dexadreson, 4 mg), atropine (Atrocare, 1 mg), and carprofen (Rimadyl, 0.15 mg) before being anesthetized with isoflurane (1.5–2%) and administered with buprenorphine (Vetergesic, 1 ml/kg) postoperatively. Mice were then placed in a stereotaxic frame (Model 900LS, David Kopf Instruments, CA, USA) and their body temperature was kept constant at 37 °C by the use of a heating mat and a DC temperature controller in conjunction with a temperature probe (FHC, ME, USA).

For injections in the auditory cortex of AAV1.hSyn.Cre.WPRE (Penn Vector Core), the skin over this part of the brain was shaved and an incision was made, after which three small holes were drilled (Foredom K.1070, Blackstone Industries, CT, USA) into the skull with a 0.4 mm drill bit and the virus injected using a pulled glass pipette and a custom pressure injection system. In order to express GCaMP6f or tdTomato in IC neurons that receive auditory cortical inputs, a total of 150–200 nl of AAV1.hSyn.Cre.WPRE was injected at three sites in the right auditory cortex of GCaMP6f (Ai95D) or tdTomato (Ai9) reporter mice, respectively, at depths of 450–550 µm below the brain surface. Given the anterograde transsynaptic spread properties of AAV1 (*Zingg et al., 2017*), this caused the expression of the desired fluorescent protein in structures that the auditory cortex projects to, including the shell of the IC (*Figure 4A and B*).

In order to prepare Gad2-Ires-Cre mice for the optogenetics experiments, we removed a large flap of skin over the parietal and temporal bones, partially removed the temporal muscles, and performed a circular craniotomy of 3 mm diameter over each auditory cortex. We then injected a total of 500 nl of AAV5-EF1a-DIO-hChR2-EYFP (UNC Vector Core) bilaterally across four sites and two depths (200 and 600 µm) into the auditory cortex. Each craniotomy was covered with a circular 3 mm glass window that was attached to the edges of the skull with cyanoacrylate glue (Pattex Ultra Gel, Henkel), and the exposed skull was sealed with dental acrylic (C&B Superbond, Sun Medical, Japan) into which a custom steel bar was embedded for head fixation. Experiments commenced approximately three weeks afterward.

The IC window implantation and cortical lesioning in the Ai95D mice were performed at least three weeks after the injections. The window implantation involved removing a flap of skin over the (inter-)parietal and occipital bone and making a circular 3 mm craniotomy over the midbrain. A 3 mm diameter glass coverslip that had been glued to a ~1 mm tall steel cylinder with 0.5 mm wall thickness was inserted into this craniotomy. The cylinder allowed us to press the glass window gently onto the brain (in order to minimize brain movement during experiments) and was then glued to the edges of the skull. For head fixation, we embedded a custom steel plate in the dental acrylic used to seal the exposed bone.

Lesions were performed as part of the cranial window implantation surgery. In those mice undergoing lesions, we removed a slightly larger flap of skin on both sides in order to expose the temporal bone, detached and deflected and/or partly removed the temporal muscle, and then made, on both sides, an elliptical craniotomy over the auditory cortex of ~3 mm (dorsoventral) by 4 mm (rostrocaudal). The exposed tissue was then aspirated (*Hong et al., 2018*) with a blunted 19 G needle connected to a suction pump (Eschmann Vp25, UK) or destroyed by thermocoagulation (*Ceballo et al., 2019*) with a cauterizer (Small Vessel Cauterizer Kit, FST, Germany) and the piece of skull that had been removed for the craniotomy was glued (Pattex Ultra Gel) back in place. In some of the lesioned mice, after completion of the imaging, 150 nl of a retrograde viral construct (rAAV2-CAG-tdTomato, UNC Vector Core) was injected into the dorsal IC across two to three sites at depths of 100–400 µm below the brain surface in order to visualize the remaining IC-projecting cortical neurons. The extent of the lesions was estimated from the histological sections and by referencing them against sections from a mouse brain atlas (*Paxinos et al., 2001*). The experimenters were not blinded to the treatment group, i.e., lesioned or non-lesioned, but they were blind to the lesion size both during the behavior experiments and most of the data processing.

In order to visualize the distribution of IC-projecting neurons in mice without cortical lesions, 150 nl of the retrograde rAAV2-CAG-cre (UNC Vector Core) construct was injected into the dorsal IC of one Ai9 mouse with an intact cortex across three sites at depths of 100–400 µm below the brain surface.

## Histology

For histological processing, mice were perfused transcardially, first with phosphate-buffered saline (PBS) and then with 4% paraformaldehyde in (PBS), and their brains were sectioned coronally (100 μm thick) with a vibratome (Leica). Images were taken manually using a Leica DMR microscope, a confocal laser scanning microscope (Olympus FV1000), or with an automated slide scanner (Zeiss Axioscan Z1). The brain of one mouse (*Figure 2—figure supplement 1*) was sectioned and imaged on a custom-built two-photon whole brain tomograph.

## Click detection task

Starting 2–3 days before training commenced, the mice were habituated to head fixation in the experimental setup, and their access to water was restricted to about ~1 ml per day, bringing their body weight down to about ~85% of the pre-restriction values. During the training phase, the mice were required to report a 0.5ms broadband click stimulus of 80 dB SPL by licking a waterspout positioned in front of them. Licking within a 1.5 s response window (occasionally this was reduced in duration to discourage excessive licking) triggered an immediate water reward (~2 μl). Stimulus trials and catch (no stimulus) trials were randomly interleaved with an inter-trial interval drawn from a normal distribution with a mean and standard deviation of 8 s and 2 s, respectively, and a lower bound of 3 s. Successful reporting of the sound within the response window was scored a 'hit,' while failure to respond was scored a 'miss.' During catch trials, neither licking ('false alarm') during the 1.5 s response window nor withholding licking ('correct rejection') triggered a reward. To help the mice form an association between sound and reward, they received occasional 'free' rewards in stimulus trials during the initial training even when no licking occurred.

Once the mice had achieved a stable level of performance (typically two days with d'>1.5), quieter stimuli (41–71 dB SPL) were introduced. For each mouse, a total of 9 different sound levels were used and the range of sound levels was adjusted to each animal's behavioral performance to avoid floor and ceiling effects and could, therefore, differ from mouse to mouse. The behavioral experiments were run using custom MATLAB (MathWorks) scripts interfacing with a National Instruments board (NI USB-6501) for reward delivery and lick registration. The stimuli were presented using Psychtoolbox through a free-field speaker (Vifa, Avisoft Bioacoustics, Germany), positioned about ~15 cm from the snout of the mouse. Stimuli were calibrated using a Pettersson M500 microphone, which was itself referenced to a sound-level calibrator (Iso-Tech SLC-1356). Stimulus levels were calibrated by integrating the recorded RMS of clicks over the mouse hearing range (1–100 kHz) and comparing this to the RMS of stimuli from the reference sound-level calibrator.

In the optogenetics experiments, the behavioral task was identical except that a single sound level (80 dB SPL) was used and on 50% of the trials bilateral photostimulation (20 Hz, 10ms pulses, 0.2 mW/mm$^2$) was performed via two 470 nm LEDs (CREE-XP-E2, LED-Tech, Germany) positioned above the cranial windows. LED-on and LED-off trials were randomly interleaved and stimulation lasted for 700 ms starting 50 ms before trial onset. Furthermore, masking flashes were presented in all trials from two bright LEDs (60 mW) positioned a few cm in front of the animals' eyes.

## Two-photon calcium imaging

Imaging was performed at a depth of 50 μm – 150 μm from the IC surface using a commercial two-photon laser-scanning microscope (B-Scope, ThorLabs, VA, USA), a SpectraPhysics Mai-Tai eHP laser (Spectra-Physics, CA, USA) tuned to 930 nm, and a Nikon 16 × 0.8 NA objective. Images were acquired with a resolution of 512 by 512 pixels at a rate of ~28 Hz. The size of the field of view was either 500 μm by 500 μm or 666 μm by 666 μm, which allowed us to, typically, image dozens of corticorecipient IC neurons simultaneously. Each imaging session lasted around 1–2 hr.

## Image processing

Rigid and non-rigid image registration, segmentation, neuropil, and signal extraction were performed using the Python version of suite2p (*Pachitariu et al., 2017*). Neuropil extraction was performed using default suite2p parameters (https://suite2p.readthedocs.io/en/latest/settings.html), neuropil correction was done using a coefficient of 0.7 and calcium ΔF/F signals were obtained by using the median over the entire fluorescence trace as F0. To remove slow fluctuations in the signal, a baseline of each neuron's entire trace was calculated by Gaussian filtering as well as minimum and maximum filtering

using default suite2p parameters. This baseline was then subtracted from the signal. To assess the extent of image displacement in the z-axis, we compared the average of the top and the bottom 500 frames of each spatial principal component (PC) of the registered images for every 8–16 min of the recordings. Any region of interest (ROI) with substantial z-axis movement was excluded from further analysis. Sessions in which the majority of ROIs had to be excluded were discarded entirely. Furthermore, in order to specifically assess brain motion caused by the motor component of the task, i.e., the animal's licking, lick-triggered movies of the imaging frames were created for every 8–16 min of the recordings. The rationale here is that if licking causes a stereotypical displacement of the imaging plane, this will become apparent when image sequences are averaged across lick events. Specifically, non-registered image sequences surrounding (from 2 s before to 2 s after) lick events were used to produce averaged lick-triggered movies. These lick-triggered movies, as well as non-averaged sequences, were then visually inspected and ROIs were excluded from subsequent analysis if they were affected by substantial z-motion.

## Analysis of task-modulated and sound-driven neurons

To identify individual neurons that produced significantly different response magnitudes in hit and miss trials, we calculated the mean activity for each stimulus trial by taking the mean activity over the 5 s following the stimulus presentation and subtracting the mean activity over the 2 s preceding the stimulus during that same trial. A Mann-Whitney U test was then performed to assess whether a neuron showed a statistically significant difference (Benjamini-Hochberg adjusted p-value of 0.05) in response magnitude between hit and miss trials. The analysis was performed using equal numbers of hit and miss trials at each sound level to ensure balanced sound level distributions. If, for a given sound level, there were more hit than miss trials, we randomly selected a sample of hit trials (without substitution) to match the sample size for the miss trials and vice versa. Sound-driven neurons were identified by comparing the mean miss trial activity before and after stimulus presentation. Specifically, we performed a Wilcoxon signed rank test to assess whether there was a statistically significant difference (Benjamini-Hochberg adjusted p-value of 0.05) between the mean activity over the 2 s preceding the stimulus and the mean activity over the 1-s period following stimulus presentation. This analysis was performed using miss trials with click intensities from 53 dB SPL to 65 dB SPL (many sessions contained very few or no miss trials for higher sound levels).

## Clustering analysis

To identify sub-populations of neurons with distinct response profiles, a clustering analysis was performed. While clustering is a useful approach for organizing and visualizing the activity of large and heterogeneous populations of neurons, we need to be mindful that, given continuous distributions of response properties, the locations of cluster boundaries can be somewhat arbitrary and/or reflect idiosyncrasies of the chosen method and thus vary from one algorithm to another. We employed an approach very similar to that described in *Namboodiri et al., 2019* because it is thought to produce stable results in high-dimensional neural data (*Hirokawa et al., 2019*). For each neuron, the trial-averaged activity was obtained by averaging across all the sound levels presented in a given session separately for hit and miss trials (given the small number of catch trials, approximately one-tenth of all trials, this analysis was restricted to stimulus trials only). Differences in the field of view size between sessions resulted in slight differences in frame rate and thus frame duration. Therefore, the activity traces were linearly interpolated to have the same number of data points (193 frames). For each neuron, the trial-averaged activity for missed trials was appended to that for hit trials, producing 386 data points per neuron for a total of 2649 neurons (n=1,697 neurons from 40 sessions with nine non-lesioned mice; n=952 neurons from 40 sessions with seven lesioned mice). To reduce the dimensionality of this dataset before applying the clustering algorithm, we performed principal components analysis (PCA) along the time axis to capture the temporal response profile for each neuron. Guided by the 'elbow' point in a scree plot visualizing the fraction of variance explained by each PC, we decided to project the dataset to the lower dimensional subspace formed by the first nine PCs.

Spectral clustering was used to cluster the resulting data. The affinity matrix was constructed by computing a graph of nearest neighbors. The hyperparameters of the clustering algorithm, including the number of nearest neighbors and the number of clusters, were optimized by a grid search to maximize the mean Silhouette Score for all samples. The Silhouette Score is a measure of the

compactness of individual clusters (intra-cluster distance) and the separation amongst clusters (inter-cluster distance). For a given sample $i$ that belongs to a cluster $C_I$, the Silhouette Score is defined as:

$$s_i = \frac{(b_i - a_i)}{max\,(a_i, b_i)}$$

where $a_i$ is the mean distance between sample $i$ and all the other samples in the same cluster, and $b_i$ is the mean distance of sample $i$ to the nearest cluster that sample $i$ is not part of. Let $C_I \vee$ and $C_J \vee$ be the number of samples belonging to clusters $C_I$ and $C_J$, and $d\,(i,j)$ be the distance between samples $i$ and $j$; $a_i$ and $b_i$ are defined as:

$$a_i = \frac{1}{C_I \vee -1} \sum_{j \in C_I, i \neq j} d\,(i,j)$$

$$b_i = \min_{J \neq I} \frac{1}{C_J \vee} \sum_{j \in C_J} d\,(i,j)$$

The resulting clusters from the hyperparameter search were further examined by plotting clusters in pairs against each other with t-distributed Stochastic Neighbor Embedding, a statistical method for visualizing high-dimensional data that involves giving each data point a location in a two or three-dimensional space (*van der Maaten and Hinton, 2008*).

## Population decoding

Logistic regression models were trained on the network activity of each session, i.e., the ΔF/F values of all ROIs in each session, to classify hit vs miss trials. This was done on a frame-by-frame basis, meaning that each time point (frame) of each session was trained separately. Rather than including all the trials in a given session, only trials of intermediate difficulty were used for the decoding analysis. More specifically, we only included trials across five sound levels, comprising the lowest sound level that exceeded a d' of 1.5 plus the two sound levels below and above that level. That ensured that differences in sound level distributions would be small, while still giving us a sufficient number of trials to perform the decoding analysis. Sessions were only included if there were at least 15 instances for both hit and miss trials. The models were trained with L2 regularization, which gave similar contributions to correlated features (i.e. individual neuronal activity) instead of discarding some of the correlated features that were also related to behaviorally-relevant information. The strength of the regularization for each model was hyperparameter-tuned and the reported results were cross-validated. Specifically, neuronal data in each session was split into five stratified folds, and each fold preserved the percentage of hit and miss trials in a given session. Four folds were used for cross-validated hyperparameter search (randomized search drawn from the log-uniform distribution between $1 \times 10^{-4}$ and $1 \times 10^{2}$), and the remaining 1 fold was used for evaluating the model after the best hyperparameters were refitted on the four folds of data. To more reliably estimate the model results, the evaluation was done for each of the five folds for each session and the average of these 5 results was taken as each session's model performance at each timepoint.

The percentage of hit and miss trials was different in each session, and the number of hit trials often exceeded the number of miss trials. To include as many trials as possible while preventing the models from taking advantage of class imbalances, balancing procedures were performed at both the model-level and the metrics-level. First, logistic regression was trained with the class weights adjusted inversely proportional to the frequency of each trial type in the training data, giving higher weights to the minority class and lower weights to the majority class. Given the total number of trials in the training data $N_T$, the number of classes $N_C$, and the number of trials for a given class $N_i$, the weight for a given class $W_i$ was defined as follows:

$$W_i = W_T/(N_C * N_i)$$

These weights were then applied to the cost function during the training process to increase the penalty for minority class misclassifications and reduce the penalty for majority class misclassifications. Second, to avoid the estimated model performance being inflated due to class imbalance, balanced accuracy (*Brodersen et al., 2010*) was used to report the model performance. Balanced accuracy was defined as the arithmetic mean of the true positive rate and the true negative rate. For a model

performing equally well on either class, the balanced accuracy is the same as the conventional accuracy (i.e. the number of correct predictions divided by the total number of predictions). However, for a model scoring above chance only because the model takes advantage of the class imbalance (i.e. consistently predicts the majority class), the balanced accuracy is at the chance level.

$$Balanced\ accuracy = \frac{1}{2} * \left( \frac{N_{true\ pos.}}{N_{true\ pos.} + N_{false\ neg.}} + \frac{N_{true\ neg.}}{N_{true\ neg.} + N_{false\ pos.}} \right).$$

Additionally, dummy models were used as baseline models to compare against the performance of the logistic regression models. Dummy models predicted the class labels (i.e. hit or miss trials) randomly while taking into account the probability of each class.

To assess whether the model performance was correlated with the number of ROIs recorded in a session, Spearman's correlation coefficient was computed between the number of ROIs in a session and the mean model performance over different 1-s time periods relative to stimulus onset (from 2 s before to 5 s after stimulus onset).

Statistical tests were conducted to compare the model performance between lesioned and non-lesioned mice, as well as between the trained models and dummy models. Since the frame rate varied slightly with the size of the field of view, the numbers of frames (193–197 frames) per 7-s trial could be different across sessions. Thus, model performance was linearly interpolated to make all sessions contain the same number of frames before statistical tests were performed at each timepoint. The model performance of each session was cross-validated and averaged across folds, and the statistical tests were performed on the distributions of the sessions' model performance. The Shapiro–Wilk test was used to determine whether a parametric or nonparametric test should be used, using $p<0.05$ as a criterion. A one-sided Wilcoxon signed-rank test or paired t-test was performed for comparing trained vs dummy models, while a one-sided Mann-Whitney U test or t-test was performed for comparing trained models for different groups of mice. Because of the smaller sample sizes, the statistical tests in *Figure 6B* were carried out after binning the scores for every two timepoints. Statistical significance was defined as $p<0.05$ after Bonferroni correction.

## Acknowledgements

We are grateful to Christopher Breen and Robert Campbell for helping with the histology, to Ben Willmore for discussing the decoding analysis and for the financial assistance provided by an Oxford-Taiwan Graduate Scholarship from the University of Oxford and the Taiwan Ministry of Education to TYL, a Wellcome 4 year PhD Studentship to YW (102372/Z/13/Z), and by a Wellcome Principal Research Fellowship to AJK (WT108369/Z/2015/Z). For the purpose of Open Access, the author has applied a CC BY public copyright licence to any Author Accepted Manuscript version arising from this submission.

## Additional information

### Competing interests

Andrew J King: Senior editor, eLife. The other authors declare that no competing interests exist.

### Funding

| Funder | Grant reference number | Author |
|---|---|---|
| University of Oxford | Oxford-Taiwan Graduate Scholarship | Tai-Ying Lee |
| Taiwan Ministry of Education | Oxford-Taiwan Graduate Scholarship | Tai-Ying Lee |
| Wellcome Trust | 10.35802/102372 | Yves Weissenberger |
| Wellcome Trust | 10.35802/108369 | Andrew J King |

| Funder | Grant reference number | Author |
|---|---|---|

The funders had no role in study design, data collection and interpretation, or the decision to submit the work for publication. For the purpose of Open Access, the authors have applied a CC BY public copyright license to any Author Accepted Manuscript version arising from this submission.

## Author contributions

Tai-Ying Lee, Data curation, Software, Formal analysis, Funding acquisition, Investigation, Visualization, Methodology, Writing – original draft, Project administration, Writing – review and editing; Yves Weissenberger, Software, Investigation, Methodology; Andrew J King, Resources, Supervision, Funding acquisition, Writing – original draft, Project administration, Writing – review and editing; Johannes C Dahmen, Conceptualization, Data curation, Software, Formal analysis, Supervision, Funding acquisition, Investigation, Visualization, Methodology, Writing – original draft, Project administration, Writing – review and editing

## Author ORCIDs

Tai-Ying Lee ⓘ https://orcid.org/0000-0001-8072-1219
Andrew J King ⓘ https://orcid.org/0000-0001-5180-7179
Johannes C Dahmen ⓘ https://orcid.org/0000-0001-9889-8303

## Ethics

All experiments were approved by the Committee on Animal Care and Ethical Review at the University of Oxford and were licensed by the UK Home Office (Animal Scientific Procedures Act, 1986, amended in 2012).

Reviewer #1 (Public review): https://doi.org/10.7554/eLife.89950.4.sa1
Reviewer #2 (Public review): https://doi.org/10.7554/eLife.89950.4.sa2
Reviewer #3 (Public review): https://doi.org/10.7554/eLife.89950.4.sa3
Author response https://doi.org/10.7554/eLife.89950.4.sa4

# Additional files

## Supplementary files

• MDAR checklist

## Data availability

Data and code associated with this study are available at https://doi.org/10.5061/dryad.b2rbnzspz and https://github.com/leetaiying/lee-et-al-2024 (copy archived at *Lee, 2024*).

The following dataset was generated:

| Author(s) | Year | Dataset title | Dataset URL | Database and Identifier |
|---|---|---|---|---|
| Lee T-Y | 2024 | Data from: Midbrain encodes sound detection behavior without auditory cortex | https://doi.org/10.5061/dryad.b2rbnzspz | Dryad Digital Repository, 10.5061/dryad.b2rbnzspz |

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
