## [Editor Report · eLife assessment]

This study demonstrates that neurons receiving inputs from auditory cortex in the inferior colliculus widely encode the outcome of a sound detection task independent of the presence of auditory cortex. This **valuable** study based on imaging of transsynaptically labelled neurons provides **convincing** evidence that auditory cortex is necessary neither for sound detection, nor to channel information related to behavioral outcome to the subcortical auditory system. This study will be of wide interest for sensory neuroscientists.

---

## [Referee Report · Reviewer #1 (Public review)]

The inferior colliculus (IC) is the central auditory system's major hub. It integrates ascending brainstem signals to provide acoustic information to the auditory thalamus. The superficial layers of the IC ("shell" IC regions as defined in the current manuscript) also receive a massive descending projection from the auditory cortex. This auditory cortico-collicular pathway has long fascinated the hearing field, as it may provide a route to funnel "high-level" cortical signals and impart behavioral salience upon an otherwise behaviorally agnostic midbrain circuit.

Accordingly, IC neurons can respond differently to the same sound depending on whether animals engage in a behavioral task (Ryan and Miller 1977; Ryan et al., 1984; Slee & David, 2015; Saderi et al., 2021; De Franceschi & Barkat, 2021). Many studies also report a rich variety of non-auditory responses in the IC, far beyond the simple acoustic responses one expects to find in a "low-level" region (Sakurai, 1990; Metzger et al., 2006; Porter et al., 2007). A tacit assumption is that the behaviorally relevant activity of IC neurons is inherited from the auditory cortico-collicular pathway. However, this assumption has never been tested, owing to two main limitations of past studies:

(1) Prior studies could not confirm if data were obtained from IC neurons that receive monosynaptic input from the auditory cortex.

(2) Many studies have tested how auditory cortical inactivation impacts IC neuron activity; the consequence of cortical silencing is sometimes quite modest. However, all prior inactivation studies were conducted in anesthetized or passively listening animals. These conditions may not fully engage the auditory cortico-collicular pathway. Moreover, the extent of cortical inactivation in prior studies was sometimes ambiguous, which complicates interpreting modest or negative results.

Here, the authors' goal is to directly test if the auditory cortex is necessary for behaviorally relevant activity in IC neurons. They conclude that surprisingly, task relevant activity in cortico-recipient IC neuron persists in absence of auditory cortico-collicular transmission. To this end, a major strength of the paper is that the authors combine a sound-detection behavior with clever approaches that unambiguously overcome the limitations of past studies.

First the authors inject a transsynaptic virus into the auditory cortex, thereby expressing a genetically encoded calcium indicator in the auditory cortex's postsynaptic targets in the IC. This powerful approach enables 2-photon Ca2+ imaging from IC neurons that unambiguously receive monosynaptic input from auditory cortex. Thus, any effect of cortical silencing should be maximally observable in this neuronal population. Second, they abrogate auditory cortico-collicular transmission using lesions of auditory cortex. This "sledgehammer" approach is arguably the most direct test of whether cortico-recipient IC neurons will continue to encode task-relevant information in absence of descending feedback. Indeed, their method circumvents the known limitations of more modern optogenetic or chemogenetic silencing, e.g. variable efficacy.

The authors have revised their manuscript and adequately addressed the major concerns. Although more in depth analyses of these rich datasets are definitely possible, the current results nevertheless stand on their own. Indeed, the work serves as a beacon to move away from the idea that cortico-collicular projections function primarily to impart behavioral relevance upon auditory midbrain neurons. This knowledge inspires a search for alternative explanations as to the role of auditory cortico-collicular synapses in behavior.

---

## [Referee Report · Reviewer #2 (Public review)]

Summary:

This study takes a new approach to studying the role of corticofugal projections from auditory cortex to inferior colliculus. The authors performed two-photon imaging of cortico-recipient IC neurons during a click detection task in mice with and without lesions of auditory cortex. In both groups of animals, they observed similar task performance and relatively small differences in the encoding of task-response variables in the IC population. They conclude that non-cortical inputs to the IC can provide substantial task-related modulation, at least when AC is absent.

Strengths:

This study provides valuable new insight into big and challenging questions around top-down modulation of activity in the IC. The approach here is novel and appears to have been executed thoughtfully. Thus, it should be of interest to the community.

Weaknesses:

Analysis of single unit activity is limited in its scope.

---

## [Referee Report · Reviewer #3 (Public review)]

Summary:

This study aims to demonstrate that cortical feedback is not necessary to signal behavioral outcome to shell neurons of the inferior colliculus during a sound detection task. The demonstration is achieved in a very clear manner by the observation of the activity of cortico-recepient neurons in animals which have received lesions of the auditory cortex. The experiment shows that neither behavior performance nor neuronal responses are significantly impacted by cortical lesions except for the case of partial lesions which seem to have a disruptive effect on behavioral outcome signaling.

Strengths:

The demonstration of the main conclusions is based on state-of-the-art, carefully controlled methods and is highly convincing. There is an in depth discussion of the different effects of auditory cortical lesions on sound detection behavior.

Weaknesses:

The description of feedback signals could be more detailed although it is difficult to achieve good temporal resolution with the calcium imaging technique necessary for targeting cortico-recipient neurons.

---

## [Author Response]

The following is the authors’ response to the previous reviews.

**Reviewer #2 (Public Review):**
Weaknesses:There are however, substantial concerns about the interpretation of the findings and limitations to the current analysis. In particular, Analysis of single unit activity is absent, making interpretation of population clusters and decoding less interpretable. These concerns should be addressed to make sure that the results can be interpreted clearly in an active field that already contains a number of confusing and possibly contradictory findings.

We addressed this important point (which was also made by reviewer #1) in our previous revision. Specifically, we included additional analyses that operate at the level of single units rather than the population level, as requested by the reviewer. For example, we assessed, separately for each recorded neuron, whether there was a statistically significant difference in the magnitude of neural activity between hit and miss trials. This approach allowed us to fully balance the numbers of hit and miss trials at each sound level that were entered into the analysis. The results revealed that a large proportion (close to 50%) of units were task modulated, i.e. had significantly different response magnitudes between hit and miss trials, and that this proportion was not significantly different between lesioned and non-lesioned mice. It is therefore no longer correct to say that “analysis of single unit activity is absent”, and we would be grateful if this statement could be changed.

**Reviewer #2 (Recommendations For The Authors):**
The authors have done a good job addressing the main concerns from the previous review. There are a few additional points that hopefully do not require substantial additional edits.Figure 5/supplements. While the authors provide compelling evidence that clusters and overall activity patterns are similar for lesioned and control animals, there do appear to be some differences. For instance, the hit/miss difference for cluster 3 (the "auditory" cluster) appears to be absent for lesioned mice (Fig 5S3 D). Can the hit-miss difference be quantified?

We agree that there are some differences between the activity profiles of lesioned and non-lesioned mice: Inspection of panels A and C of Figure 5 – figure supplement 3, for instance, indicates that there is a relatively high proportion of neurons in cluster 3 of the non-lesioned mice that exhibit prolonged elevated activity in hit trials and a relatively lower proportion of those neurons in cluster 3 of lesioned mice. This likely explains the difference in the average response profiles of cluster 3 between the two groups pointed out by the reviewer. Furthermore, there is a slightly larger pre-stimulus dip in hit trial activity for lesioned than non-lesioned mice in cluster 1, a more pronounced short latency peak in hit trial activity for lesioned mice in cluster 2 as well as differences in other clusters. However, these differences are not inconsistent with our interpretation of these data in that we describe the activity profiles as being “similar” and exhibiting a “close correspondence” (rather than as being identical). Having considered this carefully, we do not believe that attempting to quantify these small differences would add much value here or help the reader with the interpretation of these data, especially given that the activity profiles of all neurons that make up each cluster are plotted in panels A and C.

Could the mice have been using somatosensory information to perform the task? A wideband click presented from a free-field speaker could have energy in a low frequency range that triggers a whisker response. Given the moderate but not insignificant somatosensory input into the IC shell, this doesn't seem like a trivial concern, and it could substantially impact interpretation of the results. Without wanting to complicate things too much, the authors might consider one or more of these questions: What's the frequency content of the click? Can a deaf mouse perform the task? Can an AC-lesioned mouse learn/perform the task with close-field acoustic stimulation? Or for a highfrequency tone target rather than a click?

This is an interesting suggestion. We have, in the context of another study, trained mice in our lab to detect somatosensory stimulation (a brush stroke to their whiskers) and consistently found that it takes them much longer (often two weeks or more) to learn to respond to a stimulation of their whiskers than to the presentation of a sound. The brush strokes applied to the whiskers in those experiments were 50-150 ms in duration and were thus orders of magnitude greater in both their duration and amplitude and considerably more salient than any somatosensory stimulus that could potentially arise from the clicks presented here. Therefore, we consider it highly unlikely that mice learned to use somatosensory information potentially picked up by their whiskers to perform the click detection task.

L. 63. The authors might want to cite some recent work from the Apostilides lab on the properties of AC-IC projections as well as non-auditory signals in the IC.

There are two recent papers from the Apostolides lab that are relevant to our study. We already cite Quass et al., 2023. We have now added Ford et al., 2024 as well.

Changes to manuscript:

Line 81: “This raises the possibility that these context-dependent effects may be inherited from the auditory cortex (Ford et al., 2024)”.

L. 220. "sound-responsive neurons" It is possible to report the representation of sound-responsive neurons in the different clusters? This might help tease apart what processes contribute to their respective activity. Not a big problem if the samples can't be registered easily.

Sound-driven neurons were identified on the basis of a subset (those trials in which sounds were presented at levels from 53 dB SPL to 65 dB SPL) of the trials used for the clustering analysis so the analyses are not directly comparable.

p. 603. "quieter stimuli" What sound level was actually used in the 2p experiments? Was it fixed at a single level per animal?

Sound level was not fixed at a single level. A total of nine different sound levels were used per mouse. We apologize that this was not made clear previously.

Changes to manuscript:

Line 603: “Once the mice had achieved a stable level of performance (typically two days with d’ > 1.5), quieter stimuli (41-71 dB SPL) were introduced. For each mouse a total of 9 different sound levels were used and the range of sound levels was adjusted to each animal’s behavioral performance to avoid floor and ceiling effects and could, therefore, differ from mouse to mouse.”

L. 747. Something is not right with this formula. It appears that it will always reduce to a value of 1/2.

Thanks for spotting this. There are two typos in this formula. This has been fixed and now reads (line 749):Balancedaccuracy=12∗(Ntruepos.Ntruepos.+Nfalseneg.+Ntrueneg.Ntrueneg.+Nfalsepos.).